# ame-miR-34 Modulates the Larval Body Weight and Immune Response of *Apis mellifera* Workers to *Ascosphara apis* Invasion

**DOI:** 10.3390/ijms24021214

**Published:** 2023-01-07

**Authors:** Ying Wu, Yilong Guo, Xiaoxue Fan, Haodong Zhao, Yiqiong Zhang, Sijia Guo, Xin Jing, Zhitan Liu, Peilin Feng, Xiaoyu Liu, Peiyuan Zou, Qiming Li, Zhihao Na, Kuihao Zhang, Dafu Chen, Rui Guo

**Affiliations:** 1College of Animal Sciences (College of Bee Science), Fujian Agriculture and Forestry University, Fuzhou 350002, China; 2Apitherapy Research Institute of Fujian Province, Fuzhou 350002, China

**Keywords:** *Ascosphaera apis*, honey bee, *Apis mellifera*, miR-34, larva, body weight, immune response

## Abstract

MiRNAs are critical regulators of numerous physiological and pathological processes. *Ascosphaera apis* exclusively infects bee larvae and causes chalkbrood disease. However, the function and mechanism of miRNAs in the bee larval response to *A. apis* infection is poorly understood. Here, ame-miR-34, a previously predicted miRNA involved in the response of *Apis mellifera* larvae to *A. apis* invasion, was subjected to molecular validation, and overexpression and knockdown were then conducted to explore the regulatory functions of ame-miR-34 in larval body weight and immune response. Stem-loop RT-PCR and Sanger sequencing confirmed the authenticity of ame-miR-34 in the larval gut of *A. mellifera*. RT-qPCR results demonstrated that compared with that in the uninfected larval guts, the expression level of ame-miR-34 was significantly downregulated (*p* < 0.001) in the guts of *A. apis*-infected 4-, 5-, and 6-day-old larvae, indicative of the remarkable suppression of host ame-miR-34 due to *A. apis* infection. In comparison with the corresponding negative control (NC) groups, the expression level of ame-miR-34 in the larval guts in the mimic-miR-34 group was significantly upregulated (*p* < 0.001), while that in the inhibitor-miR-34 group was significantly downregulated (*p* < 0.01). Similarly, effective overexpression and knockdown of ame-miR-34 were achieved. In addition, the body weights of 5- and 6-day-old larvae were significantly increased compared with those in the mimic-NC group; the weights of 5-day-old larvae in the inhibitor-miR-34 group were significantly decreased in comparison with those in the inhibitor-NC group, while the weights of 4- and 6-day-old larvae in the inhibitor-miR-34 group were significantly increased, indicating the involvement of ame-miR-34 in modulating larval body weight. Furthermore, the expression levels of both *hsp* and *abct* in the guts of *A. apis*-infected 4-, 5-, and 6-day-old larvae were significantly upregulated after ame-miR-34 overexpression. In contrast, after ame-miR-34 knockdown, the expression levels of the aforementioned two key genes in the *A. apis*-infected 4-, 5-, and 6-day-old larval guts were significantly downregulated. Together, the results demonstrated that effective overexpression and knockdown of ame-miR-34 in both noninfected and *A. apis*-infected *A. mellifera* larval guts could be achieved by the feeding method, and ame-miR-34 exerted a regulatory function in the host immune response to *A. apis* invasion through positive regulation of the expression of *hsp* and *abct*. Our findings not only provide a valuable reference for the functional investigation of bee larval miRNAs but also reveal the regulatory role of ame-miR-34 in *A. mellifera* larval weight and immune response. Additionally, the results of this study may provide a promising molecular target for the treatment of chalkbrood disease.

## 1. Introduction

*Apis mellifera*, an excellent bee species with worldwide distribution, pollinates for a substantial quantity of crops and wildflowers [1]. In addition, *A. mellifera* is widely reared in beekeeping practice and benefits humans by producing abundant api-products such as honey and royal jelly [2]. *Ascosphaera apis* is a widespread fungal pathogen that exclusively infects bee larvae, giving rise to chalkbrood disease, which severely weakens colony population and productivity [3].

MiRNAs are endogenous small RNAs (sRNAs) ranging from 18 nt to 25 nt in length [4]. Accumulating evidence demonstrates that miRNAs are not only crucial regulators in cell metabolism, proliferation, differentiation, and apoptosis [5,6,7], but also key players in modulating a great number of pivotal biological processes such as development and immunity [8,9]. In honey bees, miRNAs have been suggested to be involved in the regulation of development [10], queen spawning [11], dance behavior [12], and immune defense [13]. A subseries of *A. mellifera* miRNAs had been documented until now, e.g., Cristino et al. [14] found that miR-932 was engaged in the formation of plasticity and long-term memory of *A. mellifera* neurons; Freitas et al. [15] reported that miR-34-5p regulated the expression of pair-rule and cytoskeleton genes in *A. mellifera* and further affected embryonic development; Michely et al. [16] discovered that both miR-12 and miR-124 exerted regulatory functions in early memory formation in *A. mellifera* workers; Liu et al. [17] observed that the expression level of ame-miR-279a in the heads of *A. mellifera* nurse bees was significantly higher than that in the heads of forager bees, overexpression of ame-miR-279a significantly decreased the proboscis extension response (PER) of workers to saccharose solution, while ame-miR-279a knockdown significantly enhanced workers’ PER. By feeding or injecting specific mimics and inhibitors, effective overexpression and knockdown of miRNAs could be achieved in a series of insects including honey bees [16,18].

MiR-34 was once called longevity-related miRNA since it played a role in promoting longevity [19]. Zhang et al. [20] reported that miR-34 promoted the apoptosis of lens epithelial cells of cataract rats via the TGF-β/Smad signaling pathway. Ye et al. [21] found that miR-34 participated in the wing polyphenism regulation of rice planthoppers. In addition, previous studies showed that miR-34 played a critical part in modulating the physiological and pathological processes of *Drosophila melanogaster*. For example, Xiong et al. [22] discovered that *D. melanogaster* miR-34 was involved in the regulation of steroid signal transduction and immune defense against bacteria, overexpression of miR-34 resulted in an enhanced host immune response, whereas miR-34 knockdown attenuated the immune response of *D. melanogaster*. Lai et al. [23] documented that miR-34 prevented chronic neurodegeneration that prevented senescence in *D. melanogaster* and suppressed development-related axonal pruning in mushroom γ neurons. However, study on miR-34 in honey bees is still lacking until now. At present, whether and how miR-34 participates in regulating the body weight and immune response of honey bees is still unclear.

Here, the expression and sequence of *A. mellifera* ame-miR-34 were verified using stem-loop RT-PCR and Sanger sequencing, followed by prediction and analysis of ame-miR-34-targeted genes; furthermore, a functional investigation of ame-miR-34 was conducted based on overexpression and knockdown to decipher the effect of ame-miR-34 on larval body weight and immune response to *A. apis* infection. The findings in this current work could not only unravel the regulatory function of ame-miR-34 in the body weight and immune response of *A. mellifera* larvae but also provide a candidate molecular target for the diagnosis and control of the chalkbrood disease.

## 2. Results

### 2.1. Molecular Verification of ame-miR-34 Expression and Sequence

Agarose gel electrophoresis showed that the fragment with the expected size (approximately 82 bp) was amplified from the 6-day-old larval gut (Figure 1A). In addition, Sanger sequencing results suggested that the fragment sequence was consistent with the predicted sequence of ame-miR-34 based on transcriptome data (Figure 1B). The results were indicative of the existence and expression of ame-miR-34 in the *A. mellifera* larval gut.

### 2.2. Expression Level of ame-miR-34 in the Larval Guts was Altered Due to A. apis Infection

Stem-loop RT-PCR results indicated that the expected fragment (approximately 150 bp) was amplified from *A. apis*-inoculated 4-, 5- and 6-day-old larval guts, whereas no fragment was amplified from corresponding uninoculated larval guts or sterile water (Figure 2A), confirming the *A. apis* infection of *A. mellifera* larvae. Additionally, RT-qPCR results demonstrated that in comparison with that in the uninfected larval guts, the expression level of ame-miR-34 was significantly downregulated (*p* < 0.001) in the guts of 4-, 5-, and 6-day-old larvae infected by *A. apis* (Figure 2B), suggesting remarkable suppression of host ame-miR-34 during the *A. apis* infection process.

### 2.3. Overexpression and Knockdown of ame-miR-34 in Uninfected and A. apis-Infected Larval Guts

The RT-qPCR results suggested that as compared with that in the mimic-NC group, the expression level of ame-miR-34 in the 4-, 5-, and 6-day-old larval guts in the mimic-miR-34 group was significantly upregulated (*p* < 0.001) (Figure 3A). Comparatively, the expression level of ame-miR-34 in the guts of 4-, 5-, and 6-day-old larvae in the inhibitor-miR-34 group was significantly downregulated in comparison with that in the inhibitor-NC group (*p* < 0.001) (Figure 3B). In addition, the expression level of ame-miR-34 in the 4-, 5-, and 6-day-old larva guts in the mimic-miR-34 group was significantly upregulated as compared with that in the inhibitor-miR-34 group (*p* < 0.001) (Figure 3C). The results together indicated that effective overexpression and knockdown of ame-miR-34 in the larval guts were achieved by feeding mimic-miR-34 and inhibitor-miR-34.

It is observed that as compared with that in the *A. apis* + mimic-NC group, the expression level of ame-miR-34 in the 4-, 5- and 6-day-old larval guts in the *A. apis*+ mimic-miR-34 group was significantly upregulated (*p* < 0.05) (Figure 4B). In contrast, the expression level of ame-miR-34 in the guts of 4-, 5-, and 6-day-old larvae in the *A. apis*+ inhibitor-miR-34 group was significantly downregulated in comparison with that in the *A. apis* + inhibitor-NC group (*p* < 0.01) (Figure 4B). Moreover, as compared with that in the inhibitor-miR-34 group, the expression level of ame-miR-34 in the guts of 5- and 6-day-old larvae in the mimic-miR-34 group was significantly upregulated (*p* < 0.01); however, the expression level of ame-miR-34 in the 4-day-old larval gut was upregulated (*p* > 0.05) but there was no significant difference between these two groups (Figure 3C). These results demonstrated that effective overexpression and knockdown of ame-miR-34 in the larval guts infected by *A. apis* were also achieved by feeding mimic-miR-34 and inhibitor-miR-34.

### 2.4. Effect of ame-miR-34 Overexpression and Knockdown on Body Weights of A. mellifera Larvae

As compared with the body weight of larvae in the mimic-NC group, that of 4-day-old larvae in the mimic-miR-34 group was increased (*p* > 0.05), whereas the body weights of 5- and 6-day-old larvae were significantly increased (*p* < 0.01) (Figure 5). In comparison with the body weight of larvae in the inhibitor-NC group, that of 6-day-old larvae in the inhibitor-miR-34 group was significantly increased (*p* < 0.001); that of 4-day-old larvae was increased (*p* > 0.05), but no significant difference was detected; while that of 5-day-old larvae was significantly decreased (*p* < 0.001) (Figure 5). Collectively, the results were suggestive of the involvement of ame-miR-34 in the modulation of the body weight of *A. mellifera* larvae.

### 2.5. Analysis of ame-miR-34-Targeted Genes and the Corresponding Regulatory Network

Ame-miR-34 was predicted to target 121 genes, including *hsp* and *abct*, and there was a complex regulatory network between ame-miR-34 and its targets (Figure 6A, see also Appendix A). GO database annotation showed that these targets were involved in six molecular function-associated terms, such as binding and catalytic activity, 10 cellular component-associated terms, such as cell and cell part, and 12 biological process-associated terms, such as cellular process and metabolic process (Figure 6B, see also Appendix A). Additionally, the target genes were also annotated to 108 KEGG pathways, such as the tricarboxylic acid cycle, biosynthesis of secondary metabolites, and Wnt signaling pathway (Figure 6C, see also Appendix A).

### 2.6. Effect of ame-miR-34 Overexpression and Knockdown on the Expression of hsp and abct in A. apis-Infected Larval Guts

The targeting relationship between ame-miR-34 and the *hsp* gene is shown in Figure 7A. The RT-qPCR results indicated that the expression level of *hsp* in the 4-, 5-, and 6-day-old larval guts in the *A. apis* + mimic-miR-34 group was significantly upregulated (*p* < 0.01) in comparison with that in the *A. apis* + mimic-NC group (Figure 7B), while that in the guts of 4-, 5-, and 6-day-old larvae in the *A. apis* + inhibitor-miR-34 group was significantly downregulated (*p* < 0.01) compared with that in the *A. apis* + inhibitor-NC group (Figure 7C).

As shown in Figure 8A, there was a potential targeting relationship between ame-miR-34 and the *abct* gene. The expression level of *abct* was upregulated (*p* > 0.05) in the 4-day-old larval gut in the *A. apis* + mimic-miR-34 group in comparison with that in the *A. apis* + mimic-NC group, while it was significantly upregulated (*p* < 0.05) in the guts of 5- and 6-day-old larvae (Figure 8B). Comparatively, as compared with that in the *A. apis* + inhibitor-NC group, the expression level of *abct* in the guts of 4-, 5-, and 6-day-old larvae in the *A. apis* + inhibitor-miR-34 group was significantly downregulated (*p* < 0.001) (Figure 8C).

## 3. Discussion

Though 259 *A. mellifera* miRNAs have currently been recorded in the miRBase database (https://www.mirbase.org/textsearch.shtml?q=apis&submit=submit, accessed on 13 April 2022), the functions of the majority of miRNAs remained to be unknown, especially those miRNAs in bee larvae. Here, molecular validation, bioinformatic analysis, and functional investigation of ame-miR-34, a DEmiRNA previously identified in *A. mellifera* larval guts responding to *A. apis* invasion, were for the first time conducted. The results of stem-loop RT-PCR and Sanger sequencing verified the existence and expression of ame-miR-34 in *A. mellifera* larval guts (Figure 1), offering a solid basis for the continuous study of the regulatory function of ame-miR-34 and the associated molecular mechanism. In addition, it was observed that the expression level of ame-miR-34 was significantly downregulated in the *A. apis*-infected 4-, 5-, and 6-day-old larval guts in comparison with that in the uninfected larval guts (Figure 2B), indicative of the participation of ame-miR-34 in the host response and the inhibition of ame-miR-34 caused by *A. apis* invasion. This implied that a certain pathway was employed by *A. apis* to suppress ame-miR-34 derived from *A. mellifera* larvae. However, additional work is needed to decipher the underlying mechanism. Previously, Zhu et al. [24] fed *A. mellifera* larvae with mimics highly expressing miR-162a in plant pollen mixed into diets and artificially raised them under laboratory conditions, the results showed that miR-162a could target *amTOR* genes of larvae, inhibit the development of larvae ovaries, delay the individual growth, and fine-tunes honey bee larva caste development. Recently, our group confirmed that effective overexpression and knockdown of ame-miR-13b [25], ame-miR-79 [26], and ame-miR-bantam [27] in the larval guts of *A. mellifera* could be achieved by feeding corresponding mimics and inhibitors. In this work, ame-miR-34 was significantly upregulated in the 4-, 5-, and 6-day-old larval guts after feeding mimic-miR-34 (Figure 3A), while significant downregulation was detected in the 4-, 5-, and 6-day-old larval guts after feeding inhibitor-miR-34 (Figure 3B); additionally, in the guts of *A. apis*-infected 4-, 5-, and 6-day-old larvae, ame-miR-34 was found to be significantly upregulated and downregulated after feeding mimic and inhibitor (Figure 4), respectively. In summary, the results suggested that ame-miR-34 overexpression and knockdown were successfully achieved in the guts of *A. mellifera* larvae without and with *A. apis* infection, further confirming the feasibility and reliability of overexpression and knockdown of miRNAs in honey bee larval guts by feeding corresponding mimics and inhibitors. These studies together provided valuable references for associated functional study with bee larval miRNAs.

Studies have shown that miRNAs were engaged in the regulation of insect physiology including body weight [28,29,30]. Fang et al. [30] found that injection of miR-982490 agomir into 5th instar *Grapholita molesta* larvae resulted in increased trehalose and triglyceride levels in hemolymph, and decreased pupation success and pupa weight. In this current work, in comparison with that in the inhibitor-NC group, the weight of 5-day-old larvae in the inhibitor-miR-34 group was significantly decreased (*p* < 0.01); comparatively, (Figure 5). The results were suggestive of the involvement of ame-miR-34 in the modulation of the body weights of *A. mellifera* larvae, which opens the door for further investigating the underlying ame-miR-34-modulated mechanism. To the best of our knowledge, this is the first report on the role of miRNAs in regulating the bee larval body weight.

In the present study, ame-miR-34 was predicted to target 121 genes (Figure 6A), which were annotated to 28 GO items relevant to molecular function (Figure 6B), cellular component, and biological process, as well as 108 KEGG pathways (Figure 6C), including a series of development and immune-associated pathways, such as the Wnt signaling pathway and endocytosis. The results demonstrated that ame-miR-34 potentially played diverse regulatory roles in the larval guts of *A. mellifera*. Two pivotal genes, the *hsp* and *abct*, were selected for further investigation. In animals and plants such as *Bacillus subtilis* [31], *Caenorhabditis elegans* [32], *A. mellifera* [33], and *Arabidopsis* [34], a number of studies have shown that the *hsp* gene is a critical regulator in maintaining cell homeostasis and protecting cells from various environmental stresses [35,36,37]. Dokladny et al. [38] found that the heat shock response and autophagy coordinate and undergo sequential activation and downregulation, which together maintain protein balance in cells. In *Drosophila*, *hsp* is engaged in the regulation of the nervous system, reproductive system, and muscles. Here, after overexpression of ame-miR-34, the target gene *hsp* was significantly upregulated in the guts of *A. apis*-infected 4-, 5-, and 6-day-old larval guts, while it was significantly downregulated after ame-miR-34 knockdown (Figure 7). These results demonstrated that ame-miR-34 was a potential positive regulator of the *hsp* gene. It was speculated that ame-miR-34 participated in the host response to *A. apis* infection via positive regulation of the expression of *hsp*.

In eukaryotes, ABC transporters can transfer compounds from the cytoplasm to the extracellular space or into organelle compartments (e.g., endoplasmic reticulum, mitochondria, peroxisome), thus playing crucial roles in cell transport and host resistance [39]. After inhibiting the expression of *MRP1* (*multidrug-resistance like protein 1*), a member of the ABC transporter family in *Drosophila*, Liu et al. [40] detected the expression level of *MRP1* in *Drosophila* gut, fat body, and Malpighian tubules, and found that the *MRP1* expression was successfully inhibited only in Malpighian tubules and the downregulation of *MRP1* in Malpighian tubules led to abnormal lipid accumulation and disruption of *Drosophila* feeding behavior and increased expression of *Hr96* (*homolog of human pregnane X receptor*). Wang et al. [41] documented that overexpression of *NlMdr49-like*, a member of the ABC transporter family in *Nilaparvata lugens*, promoted the resistance of *N. lugens* to imidacloprid, whereas *NlMdr49-like* knockdown resulted in a reduction in imidacloprid resistance. In this current work, as a putative target of ame-miR-34, the expression level of *abct* was upregulated in the 4-day-old larval gut and significantly upregulated in the 5- and 6-day-old larval guts infected by *A. apis* after overexpression of ame-miR-34; comparatively, the expression level of *abct* was significantly downregulated in the guts of *A. apis*-infected larvae after ame-miR-34 knockdown (Figure 8). This indicated that ame-miR-34 positively dominated the expression of *abct*. Notably, though the upregulation of ame-miR-34 was detected in the 4-day-old larval gut, there was no significant difference in the expression level of ame-miR-34 between *A. apis* + mimic-miR-34 group and *A. apis* + mimic-NC group, implying that other unknown factors may affect the regulation of the expression of *abct* at this day-old. However, we inferred that ame-miR-34 regulated the host response to *A. apis* infection through positive modulation of *abct* expression. More studies are required to elucidate the mechanism underlying the host response to *A. apis* challenge mediated by the ame-miR-34-*hsp*/*abct* axis. Previously, functional studies on several genes in *A. mellifera*, such as *tyr1* (*tyramine receptor 1*) [42] and *AmTsf* (*transferrin*) [43], were conducted by RNAi method. Recently, on basis of bioinformatics and RNAi method, we deciphered the molecular characteristics and biological function of *nkd* (*naked cuticle*) gene in the *A. mellfera* larval response to *A. apis* invasion [44]. In the near future, dsRNA- or siRNA-based RNAi of *hsp* and *abct* will be conducted to dissect their functions [44,45].

## 4. Materials and Methods

### 4.1. Honey Bee and Microsporidian Spore

*A. mellifera* colonies were reared in the apiary of the College of Animal Sciences (College of Bee Science), Fujian Agriculture and Forestry University. *A. apis* spores were previously isolated and kept at the Honey Bee Protection Laboratory, Fujian Agriculture and Forestry University [46,47].

### 4.2. Stem-Loop RT-PCR and Sanger Sequencing of ame-miR-34

In our previous study, based on sRNA-seq and bioinformatics, ame-miR-34 was found to be differentially expressed in *A. apis*-inoculated *A. mellifera* larval guts compared with uninoculated larval guts. Specific stem-loop primers and forward primers (F) as well as universal reverse primers (R) of ame-miR-34 (5′-TGGCAGTGTTGTTAGCTGGTTTG-3′) were designed using DNAMAN software (Table 1) and then synthesized by Sangon Biotech Co., Ltd. (Shanghai, China). Using an RNA extraction kit (Promega, Madison, WI, USA), total RNA was extracted from 4-, 5-, and 6-day-old guts of *A. apis*-inoculated larvae as well as uninoculated larvae and then divided into two portions. Next, reverse transcription was performed with stem-loop primers, and the resulting cDNA was used as a template for PCR and RT-qPCR of ame-miR-34. Following the method described by Zhu et al. [48], PCR amplification was conducted and the amplification product was then detected by 1.8% agarose gel electrophoresis. The expected fragment was extracted, ligated to the pESI-T vector (Yeasen, Shanghai, China), and transformed into *E. coli* DH5α competent cells. After microbial PCR, 1 mL of bacterial solution with a positive signal was sent to Sangon Biotech Co., Ltd. (Shanghai, China) for single-end Sanger sequencing.

### 4.3. Experimental Inoculation and Gut Sample Preparation

According to the established protocol by our group [47,49], *A. apis* spores were prepared and used for the inoculation of *A. mellifera* larvae. Briefly, (1) the previously isolated *A. apis* was cultured at 25 °C on potato dextrose agar (PDA) medium, and spores were harvested and purified following the method developed by Jensen et al. [50]; (2) the marked empty comb was placed in the three strong colonies without symptoms of chalkbrood disease, which were used as experimental colonies; (3) based on the size standard described by Brødsgaard et al. [51], 2-day-old larvae were carefully transferred to six-well culture plates, each well was preset with 800 μL of artificial diet, 40 larvae per well, and the plates were then placed in a constant temperature and humidity incubator (35 ± 0.5 °C, RH 90%); (4) 3-day-old larvae were carefully transferred to 48-well culture plates using transferring spoon; each 3-day-old larva in the treatment group was fed a diet containing 4 × 10^5^ spores/mL, while each 3-day-old larva in the control group was fed a diet without spores; the larvae that consumed all diet were used for further experiment, whereas those that did not consume all diet were discarded; 45 μL of diet was readded into each well every 12 h until 6-day-old; (5) 21 guts of 4-, 5-, and 6-day-old larvae in *A. apis*-inoculated groups and 21 guts of 4-, 5-, and 6-day-old larvae in uninoculated group were, respectively dissected according to the method reported by Guo et al. [52], immediately frozen in liquid nitrogen, and stored at −80 °C.

### 4.4. RT-qPCR Detection of ame-miR-34

Following the instructions of the Hifair^®^ qPCR SYBR Green Master Mix (Low Rox Plus) kit (Yeasen, Shanghai, China), RT-qPCR of ame-miR-34 was conducted on a QuantStudio 3 fluorescent quantitative PCR system (ABI Company, Tampa, FL, USA). *AmU6* (GenBank ID: LOC725641) was used as the internal reference. The reaction system (20 μL) included 10 μL of Hifair^®^ qPCR SYBR Green Master Mix (Low Rox Plus) (Yeasen, Shanghai, China), 1 μL of forward and reverse primers (2.5 μmol/L), 1 μL of cDNA template, and 7 μL of DEPC water. The reaction conditions were set as follows: predenaturation at 95 °C for 5 min, denaturation at 95 °C for 10 s, and annealing and extension at 60 °C for 30 s for 45 cycles; the melting curve program was the default setting of the system. The experiment was performed three times using three independent biological samples. The relative expression level of ame-miR-34 was calculated using the 2^−ΔΔC*t*^ method [53].

### 4.5. Overexpression and Knockdown of ame-miR-34 in Normal Larval Guts

According to the method described by Wang et al. [54], ame-miR-34 mimic (mimic-miR-34) and inhibitor (inhibitor-miR-34) as well as corresponding negative controls (mimic-NC and inhibitor-NC) were designed using Dharmacon (Lafayette, Colorado, USA) software and synthesized by GenePharma (Shanghai, China) (Table 2). *A. mellifera* larvae were reared following the methods developed by Peng et al. [55] with minor modifications. In brief, (1) the queen in an *A. mellifera* colony was confined to the empty comb using the queen spawning controller, and after 15 h of natural oviposition, the queen was taken out and the comb was quickly transferred to the laboratory; (2) 2-day-old larvae were carefully transferred to six-well culture plates (each well was preset with 800 μL of artificial diet), 40 larvae per well, and the plates were then placed in a constant temperature and humidity incubator (35 ± 0.5 °C, RH 90%) for 24 h; (3) 3-day-old larvae were then transferred to a 48-well culture plate, and each well was preset with 45 μL of artificial diet containing mimic-miR-34, mimic-NC, inhibitor-miR-34, or inhibitor-NC, with the final concentration of 20 pmol/g; there were four groups in total, including mimic-miR-34 group, mimic-NC group, inhibitor-miR-34, and inhibitor-NC group; (4) 45 μL of diet was readded into each well every 12 h until 6-day-old; (5) five larvae in each group were used for measurement of body weight, while three larvae were subjected to preparation of gut tissues according to our previously established method [52], followed by RNA isolation and RT-qPCR detection. There were three biological replicates for this experiment.

### 4.6. Measurement of Body Weight

Based on the method described by Borsuk et al. [56], 4-, 5-, and 6-day-old larvae (n = 5) in each group were rinsed with PBS buffer three times to remove diet residue on the body surface, followed by absorption of the fluid on the larval body surface with clean filter paper. Next, the larvae were weighed using FA2004 electronic scales (Shanghai Shunyu Hengping, Shanghai, China). There were three biological replicates for this experiment.

### 4.7. RNA Isolation, cDNA Synthesis, and RT-qPCR Detection

Total RNA from 4-, 5-, and 6-day-old larval guts was respectively extracted using an RNA extraction kit (Promega, Madison, WI, USA), and then equally divided into two portions. One portion was subjected to reverse transcription with stem-loop primers, and the resulting cDNA was used as templates for RT-qPCR detection of ame-miR-34; the other portion was reverse transcribed using Oligo DT primers, and the resulting cDNA was used as templates for RT-qPCR of ame-miR-34-targeted genes and the internal reference genes *actin* (GenBank ID: NM001185145) and *AmU6* (GenBank ID: LOC725641). RT-qPCR was performed to determine the effects of overexpression and knockdown of ame-miR-34 following the method described in Section 4.4. The experiment was performed three times using three independent biological samples. The relative expression of ame-miR-34 was calculated using the 2^−ΔΔC*t*^ method [53].

### 4.8. Overexpression and Knockdown of ame-miR-34 in A. apis-Inoculated Larval Guts

Following the method mentioned in Section 4.3, 2-day-old larvae were carefully transferred to 6-well culture plates (each well was preset with 800 μL of artificial diet), 40 larvae per well, and the plates were then placed in a constant temperature and humidity incubator (35 ± 0.5 °C, RH 90%) for 24 h; 3-day-old larvae were transferred to 48-well culture plates, each larva was fed 5 μL of *A. apis* spores at a final concentration of 4 × 10^5^/mL at 10:00 a.m. and then fed 45 μL of a diet containing mimic-miR-34, mimic-NC, inhibitor-miR-34 or inhibitor-NC (20 pmol/g). These four groups were named (1) *A. apis* + mimic-miR-34 group, (2) *A. apis*+ mimic-NC group, (3) *A. apis* + inhibitor-miR-34 group, and (4) *A. apis*+ inhibitor-NC group. Every 12 h thereafter, 45 μL of diet was added to each well. The larval guts in each of the above-mentioned groups were respectively dissected following the protocol mentioned in Section 4.5, followed by RNA isolation and RT-qPCR detection of the effects of ame-miR-34 overexpression and knockdown. There were three biological replicates for this experiment.

### 4.9. Target Prediction and Analysis of ame-miR-34

According to our previously described method [57], a combination of three software programs was employed to predict target genes of ame-miR-34, including RNAhybrid + svm_light, miRanda, and TargetScan (parameters were set as default). The intersection of the prediction results was considered the reliable set of targets. Subsequently, the target genes were mapped to the GO (https://www.omicshare.com/tools/Home/Soft/gogsea, accessed on 13 April 2022) and KEGG (https://www.omicshare.com/tools/Home/Soft/pathwaygsea, accessed on 13 April 2022) databases of *A. mellifera* by the BLAST tool with default parameters to obtain corresponding functional term and pathway annotations. Furthermore, on basis of the target prediction results, potential binding relationships between ame-miR-34 and target genes were visualized using Adobe Illustrator software (Adobe, Mountain View, CA, USA).

### 4.10. RT-qPCR Determination of ame-miR-34-Targeted Genes

Following the prediction result in Section 4.5 and associated documentation with honey bees [54,58], two target genes of ame-miR-34 were selected for RT-qPCR determination, including the heat shock protein-encoding gene *hsp* (GenBank ID: NM001160050.1) and the ABC transporter gene *abct* (GenBank ID: XM026439743). The cDNA obtained in Section 4.2 was used as templates for RT-qPCR. The *actin* gene (GenBank ID: NM001185145) was used as the internal reference. The reaction system and conditions were set as described in Section 4.4. This experiment was conducted three times with three independent biological samples. Relative expression levels of *hsp* and *abct* were calculated using the 2^−ΔΔC*t*^ method [53].

### 4.11. Statistical Data Analysis

The qPCR data were subjected to Student’s *t*-test, and *p* < 0.05 was considered statistically significant. The average body weight of larvae in each group was calculated and then analyzed by one-way ANOVA and Tukey’s post hoc test, letter mark method, followed by plotting using GraphPad Prism 8 software, *p* < 0.05 was considered statistically significant.

## 5. Conclusions

Taken together, the expression of ame-miR-34 in the larval gut of *A. mellifera* was inhibited during the *A. apis* infection process, and effective overexpression and knockdown of ame-miR-34 in the larval gut was achieved by feeding corresponding mimic and inhibitor; furthermore, ame-miR-34 participated in modulating larval body weight and potentially exerted regulatory functions in the host response to *A. apis* invasion through positive regulation of the expression of *hsp* and *abct* (Figure 9). Findings in the present study offer a valuable reference for the functional investigation of bee larval miRNAs, reveal the regulatory role of ame-miR-34 in *A. mellifera* larval weight and immune response, and provide a promising molecular target for the treatment of chalkbrood disease.

## Figures and Tables

**Figure 1 ijms-24-01214-f001:**
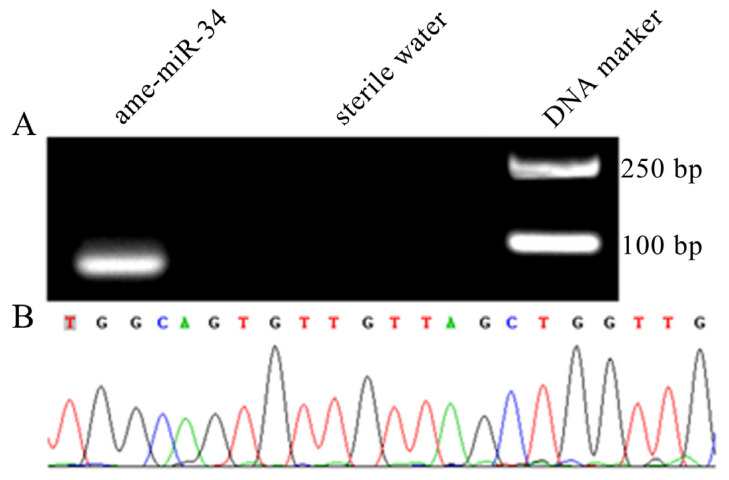
Validation of expression and sequence of ame-miR-34 in the *A. mellifera* larval gut. (**A**) Agarose gel electrophoresis for the amplification product from stem-loop RT-PCR of ame-miR-34; (**B**) Sanger sequencing of the amplified fragment from ame-miR-34.

**Figure 2 ijms-24-01214-f002:**
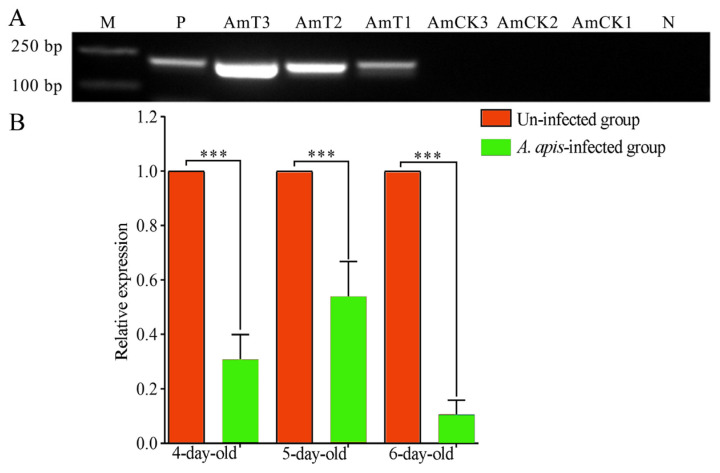
PCR validation and expression detection of ame-miR-34 in the guts of *A. mellifera* larvae inoculated with *A. apis*. (**A**) PCR validation of *A. apis* infection of *A. mellifera* larvae. Lane M: DNA marker; lane P, pure spores of *A. apis* (positive control). Lane N, sterile water (negative control); (**B**) RT-qPCR detection of host ame-miR-34 during *A. apis* infection, data were presented as mean SD and subjected to Student’s *t*-test, ns: *p* > 0.05, ***: *p* < 0.001.

**Figure 3 ijms-24-01214-f003:**
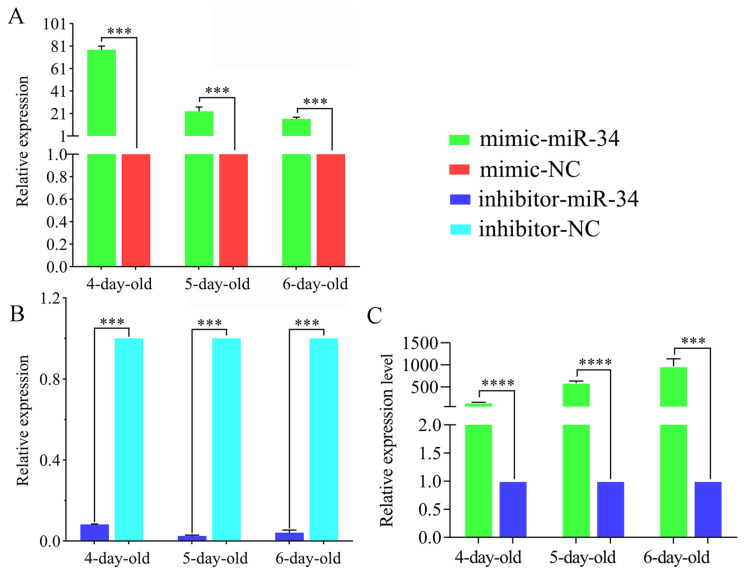
Determination of relative expression level of ame-miR-34 in un-infected *A. mellifera* larval guts. (**A**) Relative expression level of ame-miR-34 in the larval guts after feeding mimic-miR-34; (**B**) relative expression level of ame-miR-34 in the larval guts after feeding inhibitor-miR-34; (**C**) relative expression level of ame-miR-34 in the larval guts after feeding mimic-miR-34 in comparison with that after feeding inhibitor-miR-34. Data were presented as mean SD and subjected to Student’s *t*-test, ***: *p* < 0.001, ****: *p* < 0.0001.

**Figure 4 ijms-24-01214-f004:**
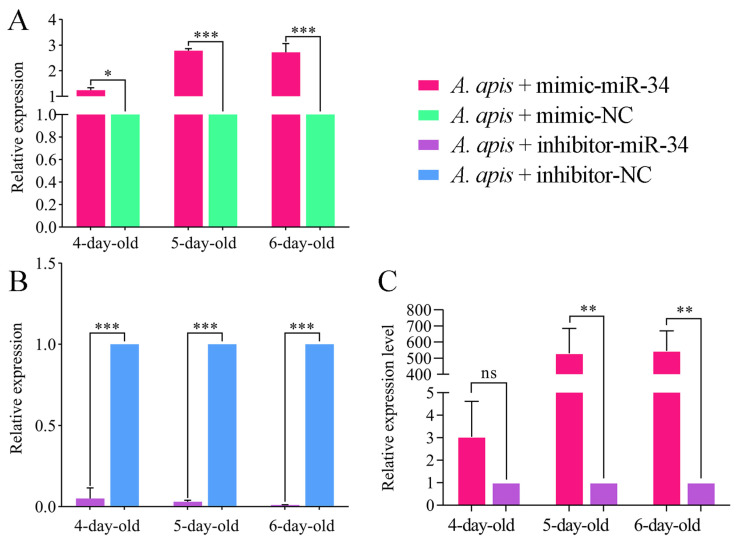
Detection of relative expression level of ame-miR-34 in *A. apis*-infected *A. mellifera* larval guts. (**A**) Relative expression level of ame-miR-34 in the *A. apis*-infected larval guts after feeding mimic-miR-34; (**B**) relative expression level of ame-miR-34 in the *A. apis*-infected larval guts after feeding inhibitor-miR-34; (**C**) relative expression level of ame-miR-34 in the larval guts after feeding mimic-miR-34 in comparison with that after feeding inhibitor-miR-34. Data were presented as mean SD and subjected to Student’s *t*-test, ns: *p* > 0.05, *: *p* < 0.05, **: *p* < 0.01, ***: *p* < 0.001.

**Figure 5 ijms-24-01214-f005:**
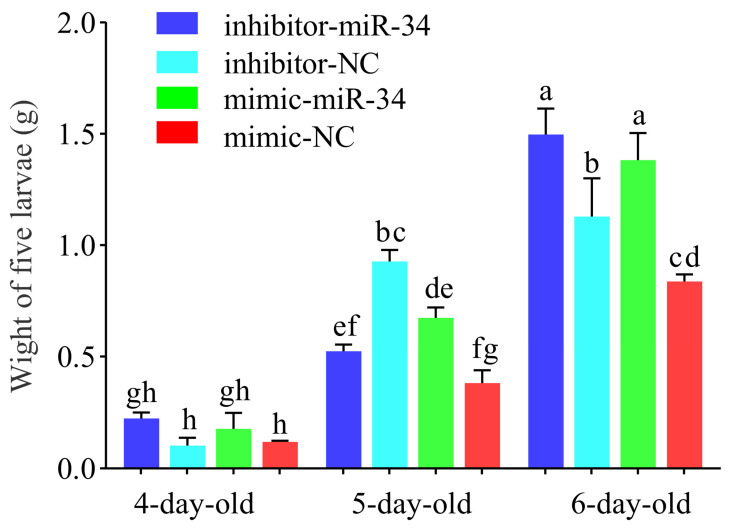
Statistics of body weights of *A. mellifera* larvae after ame-miR-34 overexpression and knockdown. SPSS statistics software was used to conduct one-way analysis of variance (ANOVA) for the average body weight of larvae in each group, with *p* < 0.05 as the significance threshold. Experimental data were compared and analyzed by the Tukey test and the letter significance labeling methods. Data are mean ± SE; the same lowercase letters above the curve indicate non-significant difference (*p* > 0.05), while the different lowercase letters above the curve indicate significant difference (*p* < 0.05) (Tukey test).

**Figure 6 ijms-24-01214-f006:**
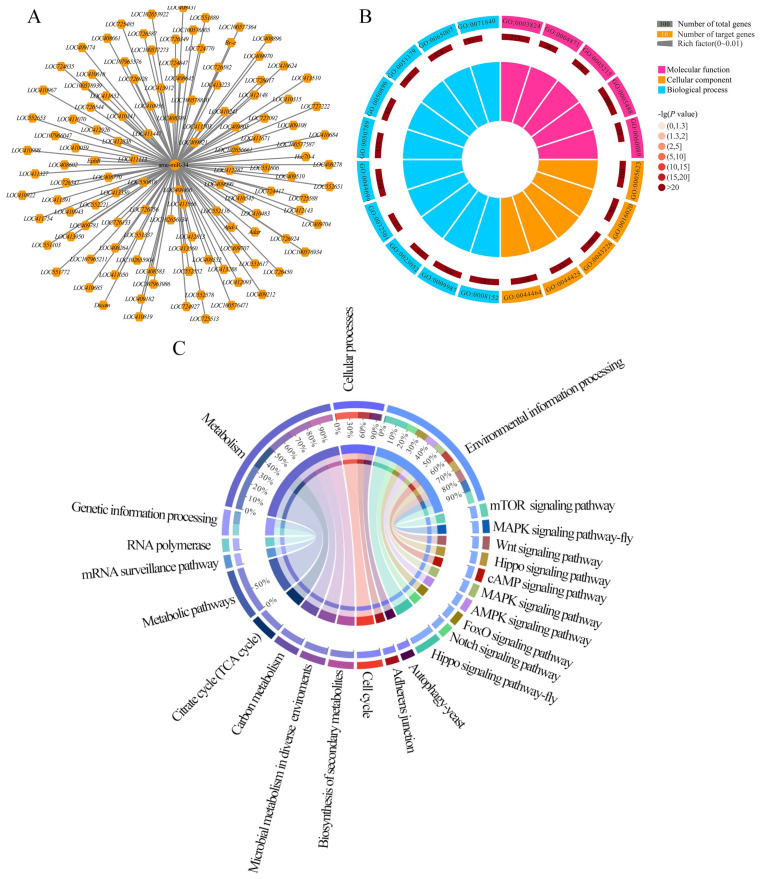
Regulatory network and database annotation of ame-miR-34-targeted genes. (**A**) Regulatory network between ame-miR-34 and target genes; (**B**) GO database annotation of targets; (**C**) KEGG database annotation of targets.

**Figure 7 ijms-24-01214-f007:**
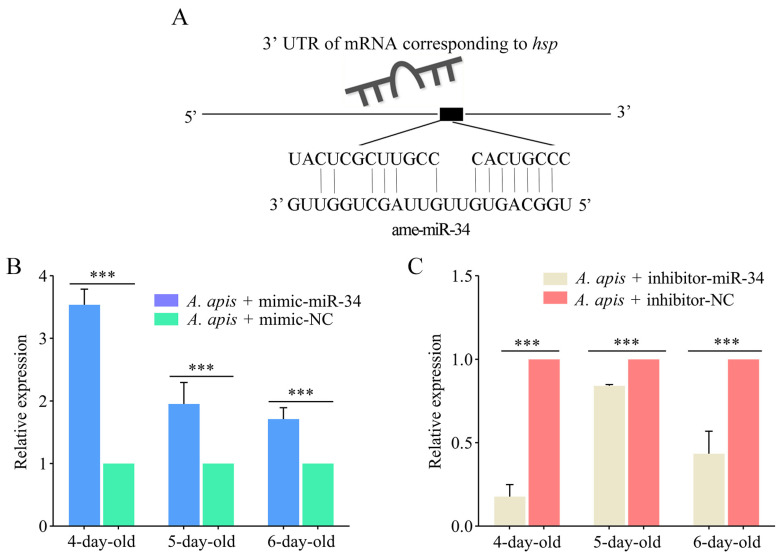
Determination of relative expression level of *hsp* in the *A. apis*-infected *A. mellifera* larval guts after ame-miR-34 overexpression and knockdown. (**A**) Target binding relationship between *hsp* and ame-miR-34; (**B**,**C**) RT-qPCR detection of *hsp* in the 4-, 5-, and 6-day-old larval guts. Data were presented as mean SD and subjected to Student’s *t*-test, ***: *p* < 0.001.

**Figure 8 ijms-24-01214-f008:**
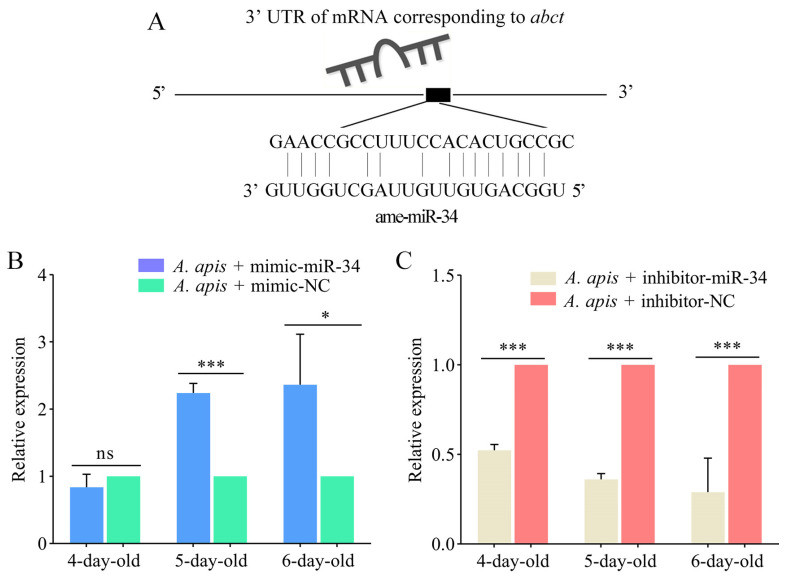
Determination of relative expression level of *abct* in the *A. apis*-infected *A. mellifera* larval guts after ame-miR-34 overexpression and knockdown. (**A**) Target binding relationship between *abct* and ame-miR-34; (**B**,**C**) RT-qPCR detection of *abct* in the 4-, 5-, and 6-day-old larval guts. Data were presented as mean SD and subjected to Student’s *t*-test, ns: *p* > 0.05, *: *p* < 0.05, ***: *p* < 0.001.

**Figure 9 ijms-24-01214-f009:**
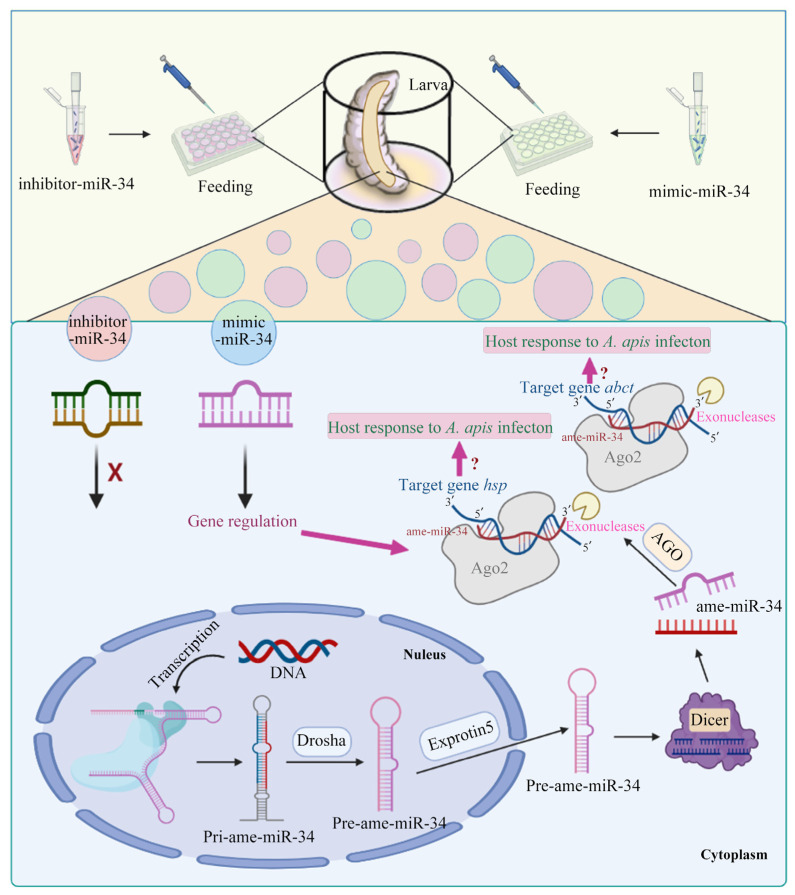
A hypothetical schematic diagram of ame-miR-34-regulated response of *A. melliferae* to *A. apis* invasion.

**Table 1 ijms-24-01214-t001:** Detailed information about primers for RT-qPCR.

Name	Sequence (5′–3′)	Purpose
ame-miR-34-F	TGGCAGTGTTGTTA	Reverse transcription of ame-miR-34
ame-miR-34-Loop	CTCAACTGGTGTCGTGGAGTCGGCAATTCAGTTGAGCAACCAGC	
ame-miR-34-R	CTCAACTGGTGTCGTGGA	
*AmU6*-F	GTTAGGCTTTGACGATTTCG	Internal reference for qPCR of ame-miR-34
*AmU6*-R	GGCATTTCTCCACCAGGTA	
*actin*-F	ATGCCAACACTGTCCTTTCTGG	Internal reference for qPCR of target genes
*actin*-R	GACCCACCAATCCATACGGA	
*hsp*-F	TCCTGTGTTGGTGTATTCCAGCATG	qPCR detection
*hsp*-R	GCAACTTGGTTCTTGGCAGCATC	
*abct*-F	ACGACGACTATACCTGGCAGTGG	
*abct*-R	CAGTTGAGACGAGACAGCATCCG	

**Table 2 ijms-24-01214-t002:** Detailed information about mimics and inhibitors used in this work.

Name	Sequence (5′–3′)	Purpose
mimic-miR-34-sense	UGGCAGUGUUGUUAGCUGGUUG	ame-miR-34 overexpression
mimic-miR-34-antisense	ACCAGCUAACAACACUGCCAUU	
inhibitor-miR-34	CAACCAGCUAACAACACUGCCA	ame-miR-34 knockdown
mimic-NC-sense	UUCUCCGAACGUGUCACGUTT	Negative control for ame-miR-34 overexpression
mimic-NC-antisense	ACGUGACACGUUCGGAGAATT	
inhibitor-NC	CAGUACUUUUGUGUAGUACAA	Negative control for ame-miR-34 knockdown

## Data Availability

Not applicable.

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
