# Peer review of "ame-miR-34 Modulates the Larval Body Weight and Immune Response of Apis mellifera Workers to Ascosphara apis Invasion"

_ijms, 2023, doi:10.3390/ijms24021214_

Round 1

Reviewer 1 Report

The manuscript studied the mechanisms of ame-miR-34 modulates the larval body weight and immune response to A. apis invasion of A. mellifera. And the results help us know the new important functions of ame-miR-34, especially against the invasion of A. apis. And the author clarified the target gene of ame-miR-34. I believe that the topic is relevant and of interest, but some corrections/suggestions should be checked.

1. Please unify the abbreviation of Apis mellifera. Both the A. mellifera and A. m. ligustica exist in the manuscript.

2. The result that effect of ame-miR-34 overexpression and knockdown on body weight was not very relevant to the research topic. The author did not go further into the reasons behind this phenomenon. So I thought author could delete the part of 2.4.

3. In the manuscript, ame-miR-34 was predicted to target 121 genes, the author screened hsp and abct genes for the further studies. Please clarified the reasons.

Author Response

Dear Reviewers,

Thanks for your comments and recommendations of great importance, which significantly improved the quality of our work. We seriously examined the whole manuscript and made corresponding modifications. All changes have been shown in red in the revised manuscript. Point-to-point responses are as follows:

Response to Reviewer 1:

  1. Please unify the abbreviation of Apis mellifera. Both the A. mellifera and A. m. ligustica exist in the manuscript.

Response: Following your helpful recommendation, we carefully checked the whole manuscript and made corresponding modifications in the revised version. Thanks.

  1. The result that effect of ame-miR-34 overexpression and knockdown on body weight was not very relevant to the research topic. The author did not go further into the reasons behind this phenomenon. So I thought author could delete the part of 2.4.

Response: Thanks for your kind comment. In this study, our major projective was to investigate the regulatory functions of ame-miR-34 in the A. mellifera larval body weight and immune response to A. apis infection, the larval body weight belonged to the physiological aspect, while the larval immune response to the pathological aspect. Though the reasons behind this phenomenon were not explored in this work, this finding reflected the regulatory potential of ame-miR-34 in the physiology of A. mellifera larvae. In addition, this finding provides an experimental basis for further study on the underlying mechanism. Hence, we retained the 2.4 section.

  1. In the manuscript, ame-miR-34 was predicted to target 121 genes, the author screened hsp and abct genes for the further studies. Please clarified the reasons.

Response: As you said, ame-miR-34 potentially targeted 121 genes in this work. The reasons why we selected hsp and abct genes for detection were: (1) a number of studies have shown that hsp gene in Bacillus subtilis (Sato, A.; Takamatsu, M.; Kobayashi, S.; Ogawa, M.; Shiwa, Y.; Watanabe, S.; Chibazakura, T.; Yoshikawa, H. Novel heat shock response mechanism mediated by the initiation nucleotide of transcription. J Gen Appl Microbiol. 2022, 68(2): 95–108.), Caenorhabditis elegans (Musa, M.; Dionisio, P.A.; Casqueiro, R.; Milosevic, I.; Raimundo, N.; Krisko, A. Lack of peroxisomal catalase affects heat shock response in Caenorhabditis elegans. Life Sci Alliance. 2022, 6(1): e202201737.), Apis mellifera (Al-Ghzawi, A.A.A.; Al-Zghoul, M.B.; Zaitoun, S.; Al-Omary, I.M.; Alahmad, N.A. Dynamics of heat shock proteins and heat shock factor expression during heat stress in daughter workers in pre-heat-treated (rapid heat hardening) Apis mellifera mother queens. J Therm Biol. 2022, 104, 103194.), and Arabidopsis (Bi, H.; Miao, J.; He, J.; Chen, Q.; Qian, J.; Li, H.; Xu, Y.; Ma, D.; Zhao, Y.; Tian, X.; et al. Characterization of the Wheat Heat Shock Factor TaHsfA2e-5D Conferring Heat and Drought Tolerance in Arabidopsis. Int J Mol Sci. 2022, 23(5): 2784.) was involved in maintaining cell homeostasis and protecting cells from various environmental stresses (Dores-Silva, P.R.; Cauvi, D.M.; Coto, A.; Silva, N.; Borges, J.C.; De Maio, A. Human heat shock cognate protein (HSC70/HSPA8) interacts with negatively charged phospholipids by a different mechanism than other HSP70s and brings HSP90 into membranes. Cell Stress Chaperones. 2021, 4, 671-684); (2) abct gene has been verified to be a pivotal player in cell transport and host resistance (Dean, M.; Rzhetsky, A.; Allikmets, R. The human ATP-binding cassette (ABC) transporter superfamily. Genome Res. 2001, 7, 1156-1166; Wang L.X.; Tao S.; Zhang Y.C.; Pei X.G.; Gao Y.; Song X.Y.; Yu, Z.T.; Gao, C.F. Overexpression of ATP-binding cassette transporter Mdr49-like confers resistance to imidacloprid in the field populations of brown planthopper, Nilaparvata lugens. Pest Manag Sci. 2022, 2, 579-590). We added the description regarding the reasons in the revised manuscript according to your helpful recommendation.

Reviewer 2 Report

The authors have presented some interesting work on an important topic. Understanding the role of microRNAs in immune function is of great importance, however, there are issues with the presentation of the methods, and important information is missing from the results and discussion. I also think that the authors’ use of t-tests instead of pairwise tests may be obfuscating some patterns that are worth commenting on.

R65: Maybe this is my own ignorance, but I’m not familiar with the term “fielder bee.” Is this the same thing as a forager bee? If so, maybe the authors should consider using this term instead as it may be more recognizable to most readers.

R71: Longevity of which organism?

L144: It would be helpful to know the significance of all treatment groups relative to each other. Would the authors please include an ANOVA with post hoc pairwise testing? It looks like the controls might be different from each other (5 days) and the treatments (6 days) are the same? If this is happening, comment on it in the discussion.

Results: Please include a section on this important topic including survivorship.

Discussion

Consider reworking the discussion to better focus on the topic. For example, everything from line 194-204 has already been mentioned in the introduction. The information could be useful, but consider working it into a greater conversation about the meaning of your results. Similarly, there is too much restating of the results.

L220: Why the overlap between MiRNA’s?

L263-264: “upregulated… but significantly upregulate…” Was is not significant in 4 day old larvae? If not, can you say that it was upregulated at all? In general, this statement is confusing, and I’m not sure what it implies.

L332: The protocol really hasn’t been described before this. I suggest including a section in the methods describing the larval rearing process before describing any methods that use rearing techniques. Did they use an established protocol? If so, please cite it. Also, the authors do not mention mortality assessments or rates in this manuscript, which is an important step in evaluating the effects of treatments.

L342: This is a lot of diet. Did the larvae consume everything they were given every 12 hours? Did you transfer them to fresh diet or was the new diet just added on top of the leftover diet?

L356: Briefly, how were the guts dissected? Include this at first mention of the procedure.

L366: This protocol is different from the above protocol where the larvae were mass reared for the first 24 hours? Why the change? How were the spores introduced to the diet? Were they homogenized in any way? Again, this is a lot of diet to consume. Were the bees transferred at any time from the 800 uL aliquot?

376: Please describe the bioinformatic analysis including all biological and computational components. Also, please include details as to how differential expression was assessed (stats?).

Author Response

Dear Reviewers,

Thanks for your comments and recommendations of great importance, which significantly improved the quality of our work. We seriously examined the whole manuscript and made corresponding modifications. All changes have been shown in red in the revised manuscript. Point-to-point responses are as follows:

  1. I also think that the authors’ use of t-tests instead of pairwise tests may be obfuscating some patterns that are worth commenting on.

Response: We seriously checked a number of documents. For the detection of the expression level of ame-miR-34 after feeding mimic and inhibitor, we think t-test may be more appropriate, since mimic was the only variate in mimic-ame-miR-34 and mimic-NC groups, while inhibitor was the only variate in inhibitor-ame-miR-34 and inhibitor-NC groups. Considering your helpful comment, we calculated the expression level of ame-miR-34 in the larval guts in the mimic-ame-miR-34 group as compared with that in the larval guts in the inhibitor-ame-miR-34 group, and the results showed that the difference was significant, as expected. For the detection of the larval body weight after feeding mimic and inhibitor, the body weight of larvae in four groups were compared using one-way ANOVA and Tukey's post hoc test, letter mark method. which would better show the difference among these groups. Thanks.

  1. R65: Maybe this is my own ignorance, but I’m not familiar with the term “fielder bee.” Is this the same thing as a forager bee? If so, maybe the authors should consider using this term instead as it may be more recognizable to most readers.

Response: Thanks for your helpful comment, following which we carefully checked the whole manuscript and replaced the term “fielder bee” with “forager bee” where necessary.

  1. R71: Longevity of which organism?

Response: The organism mentioned here is Drosophila. Thanks.

  1. L144: It would be helpful to know the significance of all treatment groups relative to each other. Would the authors please include an ANOVA with post hoc pairwise testing? It looks like the controls might be different from each other (5 days) and the treatments (6 days) are the same? If this is happening, comment on it in the discussion.

Response: Thanks for your kind recommendation. We think t-test may be more appropriate for the detection of the expression level of ame-miR-34 after feeding mimic and inhibitor, since mimic was the only variate in A. apis+mimic-ame-miR-34 and A. apis+mimic-NC groups, while inhibitor was the only variate in A. apis+inhibitor-ame-miR-34 and A. apis+inhibitor-NC groups. Considering your helpful comment, we calculated the expression level of ame-miR-34 in the larval guts in the A. apis+mimic-ame-miR-34 group in comparison with that in the larval guts in the A. apis+inhibitor-ame-miR-34 group, and added the result in the revised manuscript.

  1. Results: Please include a section on this important topic including survivorship.

Response: Thank you for your kind suggestion. Currently, our city was heavily influenced by COVID-19, our university began the winter holiday in advance, and all stuff and students cannot entre into the labs, so it’s hard for us to perform the experiment of larval rearing and survival rate detection now. We will add this result in our further study associated with ame-miR-34-regulated mechanism.

  1. Consider reworking the discussion to better focus on the topic. For example, everything from line 194-204 has already been mentioned in the introduction. The information could be useful, but consider working it into a greater conversation about the meaning of your results. Similarly, there is too much restating of the results.

Response: According to your kind suggestion, we seriously revised the discussion section in the revised manuscript and focused on the topic. Thanks.

  1. L220: Why the overlap between MiRNA’s?

Response: In Line 220, we described that “Additionally, in the guts of A. apis-infected 4-, 5-, and 6-day-old larvae, ame-miR-13b was found to be significantly upregulated and downregulated after feeding mimic and inhibitor, respectively.”, we found no content regarding the overlap between miRNAs. Thanks.

  1. L263-264: “upregulated… but significantly upregulate…” Was is not significant in 4-day old larvae? If not, can you say that it was upregulated at all? In general, this statement is confusing, and I’m not sure what it implies.

Response: Thank you so much for your professional comment. As shown in Figure 8B, in comparison with that in the A. apis+mimic-NC group, the expression level of abct was downregulated in the 4-day-old larval gut in the A. apis+mimic-miR-34 group, but there was no significance (P > 0.05) of the difference between two groups; while the expression level of abct was significantly upregulated (P < 0.05) in the guts of 5- and 6-day-old larvae. We described the object experimental results here to give more detailed information. To be more accurate, we seriously checked and modified this sentence in the revised version of manuscript.

  1. L332: The protocol really hasn’t been described before this. I suggest including a section in the methods describing the larval rearing process before describing any methods that use rearing techniques. Did they use an established protocol? If so, please cite it. Also, the authors do not mention mortality assessments or rates in this manuscript, which is an important step in evaluating the effects of treatments.

Response: Larval rearing performed in this work was based on the previously described method (Peng, C.Y.; et al. Effect of chlortetracycline of honeybee worker larvae reared in vitro. J. Invertebr. Pathol. 1992, 60: 127–133; Guo, R.; Chen. D.; Diao. Q.; Xiong, C.; Zheng. Y.; Hou, C. Transcriptomic investigation of immune responses of the Apis cerana cerana larval gut infected by Ascosphaera apis. J. Invertebr. Pathol. 2019, 166, 107210). The above-mentioned papers were cited here, and the corresponding description was improved in the revised manuscript. Currently, our city was heavily influenced by COVID-19, our university began the winter holiday in advance, and all stuff and students cannot entre into the labs, so it’s hard for us to perform the experiment of larval rearing and survival rate detection now. We will add this result in our further study associated with ame-miR-34-regulated mechanism. Thanks for your valuable recommendation and kind understanding.

  1. L342: This is a lot of diet. Did the larvae consume everything they were given every 12 hours? Did you transfer them to fresh diet or was the new diet just added on top of the leftover diet?

Response:  In this work, 2-day-old larvae were carefully transferred to six-well culture plates, each well was pre-set with 800 μL of artificial diet, 40 larvae per well. In average, 20 μL of diet were provided for each larva. Due to the small size of 2-day-old larva, 20 μL of diet could be consumed and meet the requirement. In view of that the larvae could be easily damaged if being frequently transferred using the spoon, which may seriously affect the experimental results. Here, we did not transfer them to fresh diet every 12 h, instead added fresh diet in each well. According to our observation during the whole process, the larvae almost consumed all diet (45 μL) in each well. Thanks.

  1. L356: Briefly, how were the guts dissected? Include this at first mention of the procedure.

Response: The larval guts were dissected following our previously developed method (Guo, R.; Chen. D.; Diao. Q.; Xiong, C.; Zheng. Y.; Hou, C. Transcriptomic investigation of immune responses of the Apis cerana cerana larval gut infected by Ascosphaera apis. J. Invertebr. Pathol. 2019, 166, 107210). We added a brief introduction of the larval gut dissection method and cited the above-mentioned reference in the revised manuscript. Thanks.

  1. L366: This protocol is different from the above protocol where the larvae were mass reared for the first 24 hours? Why the change? How were the spores introduced to the diet? Were they homogenized in any way? Again, this is a lot of diet to consume. Were the bees transferred at any time from the 800 uL aliquot?

Response: Thank you for your valuable comment, following which we seriously examined the corresponding description. This is a mistake here. In fact, the protocol described here was the same as that described in the 4.5 section. Accordingly, we made necessary modifications in the revised version of manuscript.

  1. 376: Please describe the bioinformatic analysis including all biological and computational components. Also, please include details as to how differential expression was assessed (stats?).

Response: Following your kind suggestion, detailed information about bioinformatic analyses and differential expression investigation were added in the methods section in the revised version of manuscript. Thanks.